# qc3C: Reference-free quality control for Hi-C sequencing data

**Matthew Z. DeMaere** *, **Aaron E. Darling**

The iThree Institute, University of Technology Sydney, Ultimo, NSW, Australia

* matthew.demaere@uts.edu.au

## Abstract

Hi-C is a sample preparation method that enables high-throughput sequencing to capture genome-wide spatial interactions between DNA molecules. The technique has been successfully applied to solve challenging problems such as 3D structural analysis of chromatin, scaffolding of large genome assemblies and more recently the accurate resolution of metagenome-assembled genomes (MAGs). Despite continued refinements, however, preparing a Hi-C library remains a complex laboratory protocol. To avoid costly failures and maximise the odds of successful outcomes, diligent quality management is recommended. Current wet-lab methods provide only a crude assay of Hi-C library quality, while key post-sequencing quality indicators used have—thus far—relied upon reference-based read-mapping. When a reference is accessible, this reliance introduces a concern for quality, where an incomplete or inexact reference skews the resulting quality indicators. We propose a new, reference-free approach that infers the total fraction of read-pairs that are a product of proximity ligation. This quantification of Hi-C library quality requires only a modest amount of sequencing data and is independent of other application-specific criteria. The algorithm builds upon the observation that proximity ligation events are likely to create *k*-mers that would not naturally occur in the sample. Our software tool (qc3C) is to our knowledge the first to implement a reference-free Hi-C QC tool, and also provides reference-based QC, enabling Hi-C to be more easily applied to non-model organisms and environmental samples. We characterise the accuracy of the new algorithm on simulated and real datasets and compare it to reference-based methods.

**Data Availability Statement:** qc3C has been implemented in Python3 for the Linux operating system. The project has been released under GNU AGPLv3 and hosted on Github. https://github.com/cerebis/qc3C Source code, execution environment, and analysis results used in preparing this

## Author summary

The Hi-C sequencing technique offers the potential for significant scientific insight about the spatial arrangement of DNA, however achieving such outcomes is highly dependent on the quality of the resulting sequencing library. Unlike conventional next-gen sequencing, only a fraction of a given Hi-C library contains this useful spatial information (the signal) with the remainder being effectively noise. As Hi-C remains a challenging laboratory technique, signal strength of resulting libraries can vary greatly. As a quality metric, the quantification a library's signal content is an essential asset in any quality mitigation strategy. Quality assessment of Hi-C data has until now relied on access to a (ideally refined)

manuscript have been deposited on Zenodo. https://doi.org/10.5281/zenodo.5099549.

**Funding:** This research was supported by the Australian Government through the Australian Research Council Discovery Projects funding scheme under the project DP180101506, http://purl.org/au-research/grants/arc/DP180101506 (to AED). The funders had no role in study design, data collection and analysis, decision to publish, or preparation of the manuscript.

**Competing interests:** The authors have declared that no competing interests exist.

reference sequence, by which indirect indicators of quality are determined. Here we describe qc3C, a software tool capable of the direct, reference-free estimation of the signal content of a Hi-C library. In doing so, not only can researchers make informed decisions on how to progress based on library information content, but eliminating the reference also enables Hi-C quality management for non-model organism and metagenomics researchers.

This is a *PLOS Computational Biology* Software paper.

## Introduction

Hi-C is a next-generation sequencing (NGS) technique that captures genome-wide evidence of spatially interacting DNA loci, where these interactions may be intra- or inter-molecular in nature. Originally developed for the 3D structural analysis of human chromatin [1], the technique has since been successfully applied to other challenging problems in genomics, such as the scaffolding of large genome assemblies, haplotype phasing, and the accurate resolution of metagenome-assembled genomes (MAGs) [2–6]. These diverse applications have demonstrated the value of Hi-C to an increasingly wide audience and a fast-growing number of researchers are utilizing the approach to answer questions of interest.

While substantial new insights can be procured from a Hi-C experiment, the library generation protocol is complex; imposing greater time-commitment, cost, and sample requirement than conventional shotgun library preparation for Illumina paired-end sequencing. The added complexity introduces new error sources and failure points, which in the face of a more time-consuming and costly experiment, is of real concern. To that end, optimisation efforts from both the commercial and academic spheres have gone in search of more robust and simplified protocols [7–9]. While such work has led to reduced time-commitment and consistently better data quality, the routine production of high-quality Hi-C data is still a greater challenge than with conventional sequencing. The risk of negative outcomes can be mitigated, however, by adopting a quality management strategy for Hi-C data production and subjecting each experiment to quality control (QC). In broad terms this involves performing QC at several stages of a developing Hi-C library (pre- and post-sequencing), which in turn permits user interventions that range from small tweaks to abandonment [7]. Although this fine-grained approach to improving the odds of success presents additional work, it is also particularly effective at minimising the cost of failure by avoiding the worst case scenario of deeply sequencing a failed library.

Methods for assessing quality at different stages of the Hi-C protocol have previously been devised [7, 9–11] and though implementing an overall quality management strategy is already possible, post-sequencing quality assessment warrants further improvement. Much of the collective time spent on software development within the field of Hi-C has been focused on new applications, the refinement of existing methods, or the implementation of end-to-end pipelines. This work has expanded the scope of Hi-C, lowered the barrier to entry, and brought about workflow standardisation, however, there has been a lack of time invested in developing methods to assess Hi-C library quality from sequencing data. Though post-sequencing would seem too late to benefit from user intervention, it is also the most definitive stage at which to

ascertain library quality. A full-scale sequencing run is also not required, as reliable QC results can be obtained from a pilot-scale sequencing run with minimal cost and sample consumption.

As a somewhat secondary consideration, current post-sequencing Hi-C statistical quality indicators are the end product of many simple criteria, whose collective aim is primarily application orientated (e.g. producing a min-noise/max-signal contact matrix). As such, these indirect indicators only imply Hi-C signal content. In contrast, our reference-free method attempts to estimate the entire fraction of Hi-C pairs, regardless of the separation distance for any individual pair or whether it would provide useful contact information.

The calculation of current indicators also requires a reference genome, which ideally should be highly contiguous and derive from the same genomic source as the Hi-C library under scrutiny. Although the time and cost investment to construct such a reference has been made for several model organisms, this is not the case in genomics of non-model organisms and in metagenomics. When undertaking these latter project types, a reference may not be immediately available and—if Hi-C quality management is desired—forces the production of an adequate quality reference first. For non-model organisms with large genomes, the challenge and cost to produce a high quality draft assembly can be significant. Eliminating the need for a reference in Hi-C QC would eliminate this imposed ordering of sequencing tasks. In doing so, sequencing projects are freed to generate and QC shotgun and Hi-C data in whatever order they happen to occur. Further, the need for a reference brings forth logistical considerations. Production environments found in busy sequencing facilities, whose sequencers analyse a wide array of genetic material, benefit from a minimum of runtime complications. We expect that the elimination of per-project references would be a welcome simplification. Lastly, reference quality will no longer have any bearing on QC results.

We expect that devising a new Hi-C quality measure which overcomes the shortcomings of existing methods and distilling the approach into a purpose-built tool would be of value to the genomics research community. We have implemented such a reference-free approach as qc3C, an easy to use and open source tool for Hi-C library quality assessment (https://github.com/cerebis/qc3C). Additionally, qc3C provides various reference-based quality measures and integrates with the MultiQC reporting framework [12]. We characterise the accuracy of the new algorithm on simulated and real datasets and compare the results to reference-based methods (S1 Fig).

## Background

### Signal and noise

For the standard Hi-C library protocol, observations of DNA:DNA interactions are captured through an ingenious sample preparation method beginning with intact cells or tissue. Briefly, the DNA within the sample is first cross-linked in vivo using formalin fixation, which freezes in place the physical contacts of DNA and proteins in each cell. The cells are then lysed and the cross-linked DNA-protein complexes are extracted and purified. The purified DNA-protein complexes are then restriction digested to create free-ends, which are then biotinylated and blunted. The DNA-protein complexes are placed under dilute conditions or immobilised on a solid substrate, followed by free-end ligation. The environment created by diluting or immobilising the complexes strongly biases the ligation reaction to join free-ends that were spatially nearby in vivo—an event known as a proximity ligation (PL). After the ligation step is completed, the cross-linking is reversed and proteins digested. The DNA is then sheared and biotin affinity purification is used to enrich for fragments containing biotin tagged PL

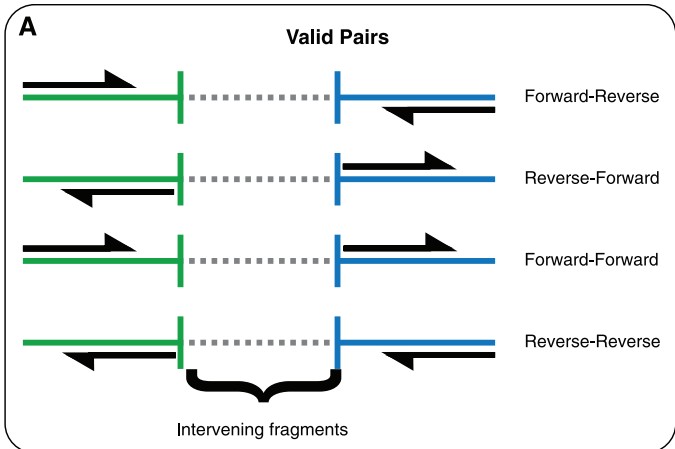

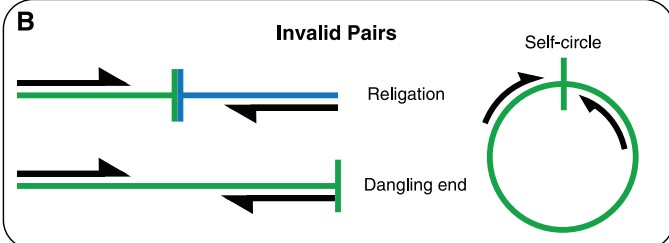

**Fig 1. Hi-C read-pair classifications.** A reference-based classification scheme for read-pairs generated by the Hi-C protocol has previously been devised [11, 13]. Regardless of orientation, read-pairs are considered valid if two or more intervening enzymatic cut-site exist between them. Conversely, invalid pairs have one or zero intervening cut-sites. No additional evidence of ligation is involved. Recent progress that has increased the resolution of the Hi-C protocol has created challenges to the interpretation of these classes. Further invalid cases exist which are not shown, these include cases of unpaired reads and when one or both ends of a pair do not map to the reference.

junctions. Finally, the PL enriched fragments become template DNA for an Illumina paired-end sequencing library.

When a Hi-C library is sequenced, the fraction of paired reads that derive from PL-containing fragments constitute the signal within the experiment, while the remainder constitute the noise and are the product of failure-modes within the Hi-C protocol (Fig 1). Examples of possible failure modes include: non-specific binding during purification, pull-down of biotinylated free ends that were not ligated, and spurious ligation products. For those pairs which form the signal, each read in the pair corresponds to one of the two interacting loci (Fig 2).

In relative terms, the signal content (or efficiency) of a Hi-C library is defined as $n_{PL}/N$, where $n_{PL}$ is the number of PL-containing read-pairs and $N$ the total number of pairs generated from the library during a sequencing run. Since algorithms that exploit Hi-C sequencing data are fundamentally counting experiments, their statistical power is strongly influenced by the available number of PL observations. As such, signal content is quite possibly the best indicator of Hi-C library quality, but the direct calculation of $n_{PL}$ is infeasible as there is no intrinsic identifying feature present across all pairs.

## Current measures of Hi-C library quality

Two reference-based statistics in widespread use as measures of library quality are the fraction of long-range pairs and the fraction of valid pairs.

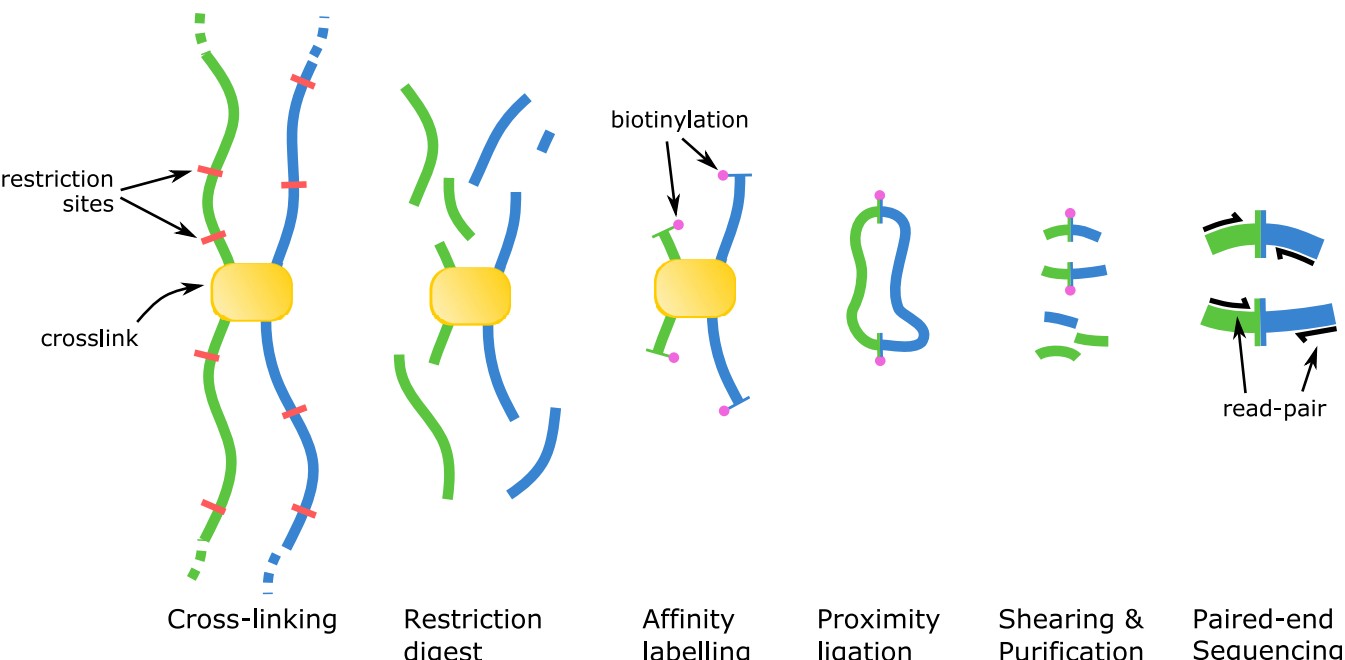

**Fig 2. Major Hi-C library protocol steps.** An outline of the steps involved in the Hi-C library protocol. From left to right, the protocol begins by cross-linking DNA in situ, after which cells are lysed and the DNA is restriction digested. Next, free ends are affinity labelled with biotin and made blunt. Next, in dilute conditions or solid substrate immobilised, the free ends are ligated, forming biotin labelled proximity ligation junctions. Next, cross-linking is reversed and bound proteins digested, followed by DNA shearing and purification. The resulting DNA fragments, enriched for proximity ligation junctions, become template DNA for an Illumina paired-end library.

Long-range pairs are those pairs that, when mapped to a reference, are found to be separated by a distance greater than can be explained by the expected shotgun library insert length (i.e. >1000 bp). This statistic only considers intramolecular (cis) mapping pairs and does not explicitly identify true PL-containing pairs. Rather, it relies on the implication that large separation distance requires a library fragment that was produced by proximity ligation and assumes no possibility of genomic rearrangement or mapping error.

Valid pairs are a conceptual classification based on how free-ends are believed to interact during the PL reaction [11, 13]. Here, free-ends that go on to form informative (valid) PL products must derive from different, non-adjacent restriction fragments produced during enzymatic digestion (Fig 1A). While intermolecular mapping pairs are inherently non-adjacent, testing for intramolecular non-adjacency requires that paired reads map to restriction fragments separated by at least one other fragment (two intervening cut-sites). Conversely, invalid pairs are those whose ends map to the same fragment, which include: unligated dangling ends (when ends face outward) and self-circles (when ends face inward) (Fig 1B). When inward facing ends map to different fragments that face the same restriction site, it is termed religation. Although formed through PL, religations are uninformative and therefore undesirable. As with long-range pairs, meeting the test of validity does not guarantee that a pair derives from PL reaction. In particular, as the step of restriction digest has evolved to use 4-cutters and dual-enzyme digests, the typical length of restriction fragments has become very small ($\approx$ 100 bp), making this criterion of little value to discriminate shotgun fragments from those generated by PL.

## Reference-free estimation of Hi-C library quality

We propose an alternative approach that identifies specific intrinsic sequence motifs created by the PL reaction in order to estimate the fraction of Hi-C read-pairs in a sample. By searching read data for these motifs and compensating for the portion of library fragments that remain unsequenced using a simple statistical model, the fraction of Hi-C read-pairs can be inferred without the need for a reference.

Restriction endonucleases used in the Hi-C protocol produce overhangs, which are then end-filled with biotinylated nucleotides. When free-ends are joined in the subsequent ligation step, the junction contains a copy of the overhang sequence from each end. For single enzyme digests this creates a sequence motif that is a duplication of the overhang sequence, while for dual enzyme digests there is a range of sequence motifs generated. In either case, a ligation artefact (LA) is introduced and marks the junction point of the PL fragment (Fig 3). The later step of random shearing acts to uniformly distribute the occurrence of the PL junction across the fragments in which they are contained. During sequencing, each paired read has the potential to pass through the junction. Termed read-through, this event results in chimeric reads that when mapped to the reference can sometimes be identified as non-overlapping split alignments. The odds of observing read-through depend upon read length and distribution of library fragment sizes. When the length of individual fragments exceeds two-fold read length, an unsequenced region begins to appear in which read-through cannot not be observed.

Though the junction sequence is easy to predict for a given enzyme, its short length means that simply counting occurrences within a readset to estimate Hi-C signal content would lead to a high false-discovery rate (FDR). Instead, we build upon the observation that, being introduced, LA are likely to create new $k$-mers that are not naturally occurring in the sample and as such should be rare in comparison to the surrounding native $k$-mers. Our algorithm (S2 Fig)

| 1. Native DNA | 5'-XXXXGATCYYYY-3' <br> 3'-xxxxCTAGyyyy-5' |
|---|---|
| 2. Restriction digest | 5'-XXXX GATCYYYY-3' <br> 3'-xxxxCTAG yyyy-5' |
| 3. End-fill and re-ligation | 5'-XXXXGATCGATCYYYY-3' <br> 3'-xxxxCTAGCTAGyyyy-5' |

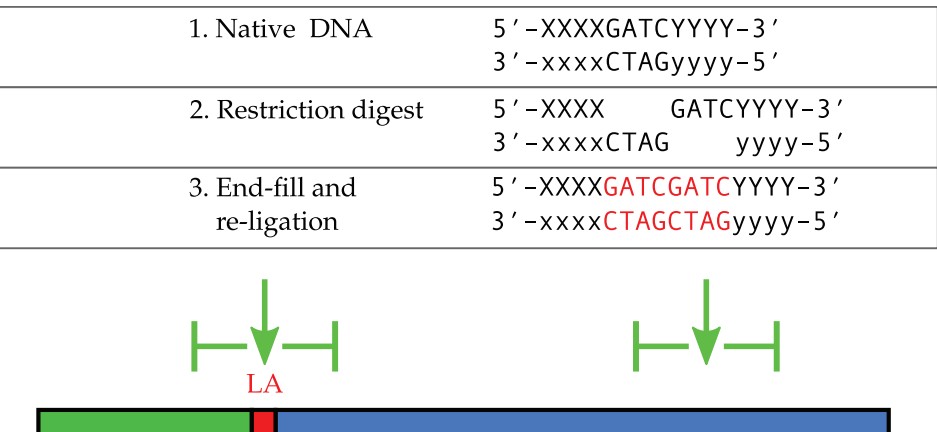

**Fig 3. Leveraging Hi-C ligation artefacts.** In Hi-C sequencing data, ligation artefacts (LA) are small sequence duplications produced by the processes of end-fill and ligation. The joining of free-ends results in chimeric DNA fragments (stylised as the thick green and blue horizontal bars). At the ligation site (intervening red bar) a ligation artefact (LA) with a predictable sequence is created. Our reference-free method to estimate Hi-C signal content exploits this introduced sequence. By sampling the frequency of local $k$-mers on either side (small grey bars) and at the junction site (small red bars), we assume the relative local frequency $\phi_i^\star$ at LA sites is characteristically lower than at any native positions $\phi_i$.

uses an empirical cumulative distribution of relative $k$-mer frequency to estimate the probability that a read containing a predicted LA was generated through the PL reaction, which in turn allows the total fraction of Hi-C reads to be estimated.

## Method

### Construct $k$-mer database

To query the frequency of arbitrary $k$-mers, $k$ is first chosen and a frequency database is built from the supplied Hi-C readset using the $k$-mer counter Jellyfish [14].

As we intend to query the frequency of local $k$-mers flanking a LA identified within a read, for the sake of specificity, it is necessary that $k$ be substantially larger than $l_{LA}$, where $l_{LA}$ is the LA length. For modern Hi-C protocols in common use $l_{LA} = 8$, while for the older 6-cutter protocol based on HindIII $l_{LA} = 10$. The size of $k$, however, also defines the total width of the surrounding region and limits how close we can approach either end of a read and still carry out our local $k$-mer frequency inspection. A trade-off, therefore, exists between increased specificity of local $k$-mers and decreased sensitivity due to the resulting unobserverable region of each read. In testing, we have found values between $24 \leq k \leq 32$, to be similar in their predictive performance. A default of $k = 24$ strikes a balance between these two competing objectives, while also producing smaller databases better suited to small-memory, small-storage computers.

### Extract local $k$-mers

Next, a random sample of paired reads from the same readset is taken and each read is checked for the presence of the LA sequence. If a LA is present in the $i^{th}$ read, its starting location is defined as $x_i$, otherwise $x_i$ is drawn uniformly at random along the read $x_i \sim U(k, l_{read} - l_{LA})$. A set of six flanking $k$-mers are extracted from either side of the LA site, three from the left $(x_i - (k + j), j \in [-1, 0, 1])$ and three from the right $(x_i + l_{LA} + j, j \in [-1, 0, 1])$, which together form the `outer` set. Next, three $k$-mers centered on the LA site are extracted $(x_i - j, j \in [-1, 0, 1])$, which together form the `inner` set.

### Calculate relative frequency

The frequencies of the nine $k$-mers are then queried from the database. If there are any missing $k$-mers the calculation is abandoned, where missing $k$-mers are a possibility if a minimum acceptable quality threshold is applied during the database generation. The relative frequency $\phi_i$ about $x_i$ is calculated as the ratio of geometric means of the `inner` to `outer` sets ($\phi_i = $ GM (`inner`)/GM(`outer`)). The calculation is abandoned, however, if $GM(outer) < \overline{f}/2.5$, where $\overline{f}$ is the mean frequency of the supplied $k$-mer database. Here, the rationale is that the naturally occurring $k$-mers in `outer` should have sufficient coverage, but this constraint is not imposed on the `inner` $k$-mers.

### Bootstrap confidence interval

Following the above procedure and iterating over a set of reads, a table of relative frequency observations is constructed ($T = (\phi_i, \lambda_i)$), where $\lambda_i$ is a binary indicator variable for the presence/absence of the LA sequence (present $\rightarrow \lambda_i = 1$).

Bootstrap sampling of this table permits a confidence interval to be estimated for the fraction of observed LAs which were produced through the proximity ligation reaction (true LAs) as follows. A random sample $t_n$ is taken from $T$ with replacement and equal cardinality. From this sample, the collection of relative frequencies $\phi_i$ for observations which did not contain the

LA ($\lambda_i = 0$) are selected and reduced to a unique set of values (duplicate observations removed). The set of unique frequencies is placed in ascending order and empirical p-values then assigned by rank-order ($\hat{p}(\phi) = (r(\phi) + 1)/(n + 1)$, where $r(\phi)$ is the number of observations $\leq \phi$) to produce a lookup table mapping relative frequency $\phi_i$ to empirical p-value [15]. Next, these p-values are distributed over the observations which do contain the LA ($\lambda_i = 1$), by finding the nearest but not greater frequency within the lookup table. The p-values of the LA containing observations are then summed to provide a sample estimate of the fraction of true LA events $\rho_n$. Lastly, from the total pool of sample estimates $\rho_n$ (default: 100 bootstrap samples), the quantiles $q = [0.025, 0.975]$ are used to determine a 95% confidence interval (CI) for the proportion of observed junction sequences considered to be the product of proximity ligation and by implication the raw proportion of observed valid Hi-C read-pairs $\rho_{obs}$.

### Correction for the unobserved

In paired-end sequencing, when an insert is longer than twice the read length there is a region centered at the insert midpoint which goes unsequenced and consequently unobserved. When testing the reads of a library for the presence of LA, the unobserved regions of these long inserts accumulate. If not compensated for, the accumulated unobserved extent leads to a systematic underestimation of the total proportion of Hi-C pairs ($\rho_{lib}$). Conversely, for inserts shorter than twice read length there is overlap centered at the insert midpoint. When reads from such short inserts are inspected independently, we must instead compensate for the possibility of duplicate observations. Further, since we require a full sequence match for the positive identification of an LA, there is an additional unobserved region at the $3'$ end of every read equal to $l_{LA} - 1$. Lastly, qc3C makes the simplifying assumption that the occurrence of LA along an insert is uniformly distributed ($x_{LA} \sim U(l_{insert})$).

Alternatively, pre-processing can be applied to pairs with sufficient overlap; merging them into a single sequence. As a consequence, the complete insert sequence of such merged pairs can be easily inspected and the possibility of duplicate observations eliminated. However, since a library is unlikely to be composed entirely of "merge-able" read-pairs, pre-processed libraries require the simultaneous handling of both merged and unmerged pairs.

Taking these considerations in hand and given a user-supplied estimate of mean insert length, an expected value for the unobserved fraction ($\alpha_{lib}$) is computed as follows. For each sample $t_n$ taken in the bootstrap sampling procedure outlined above, the unobserved fraction for the $i^{th}$ read-pair $\alpha_i$ is calculated as follows. For merged pairs, although the entire insert is present, no LA sequence can be confirmed beyond the coordinates $l_i - l_{LA}$ and therefore $\alpha_i = 1 - (l_i - l_{LA})/l_i$. For unmerged pairs, a binary coverage mask is computed using pair lengths ($l_i^1, l_i^2$) against the user-supplied expected insert length $\overline{L}$, while also noting the inaccessible region in each read as before. After computing the mask, the mean occupancy is used to infer the unobserved fraction as $\alpha_i = 1 - |\text{mask}(l_i^1, l_i^2, \overline{L})|$. The unobserved fraction of the sample $t_n$ is then calculated as the arithmetic mean $\alpha_n = |\alpha_i|$. From the pool of sample estimates $\alpha_n$, the quantiles $q = [0.025, 0.975]$ are used to determine a 95% CI for the unobserved fraction of the library $\alpha_{lib}$. Finally, the estimated fraction of Hi-C read-pairs for the whole library is estimated as $\rho_{lib} = \rho_{obs}/(1 - \alpha_{lib})$.

## Results

### Simulated data

To validate qc3C's reference-free method, a simulated parameter sweep was implemented in Nextflow [16], where reads were generated using the Hi-C read simulator sim3C (v0.3.2_py3)

[17] with E.coli MG1655 (acc: GCF_000005845.2) as the genomic source. To better match the simulated Hi-C reads with those of the read data sets, a custom 150 bp machine profile was generated using 70.1 M reads sampled across all accessions from [7]. Three parameters were varied within the sweep: the fraction of Hi-C pairs (0.01, 0.05, 0.1, 0.25, 0.5, 0.75), number of generated pairs (12500, 25000, 50000, 100000, 250000, 500000), and mean insert length (150, 200, 250, 300, 350, 400, 500 bp). Twenty randomly drawn seed values were used to produce twenty replicates for each point in the sweep and to initialise the random state of qc3C in each case. Lastly, read length was fixed at 150 bp and DpnII was chosen as the restriction enzyme.

For each of the 5,040 simulated libraries, both a "cleaned" and a "cleaned and merged" readset was produced using fastp with default parameters (version: 0.20.1) [18]. Based on sequence overlap, a significant fraction of read-pairs were merged for shorter insert lengths. Subsequently, 10,080 reference-free estimates of Hi-C signal content were computed with qc3C (5,040 merged libraries, 5,040 unmerged libraries). In each instance, all available read-pairs were used to both construct the *k*-mer library and estimate signal content.

To consider the accuracy and precision of reference-free estimation over the sweep, the relative deviation of predicted ($p_i$) from actual ($a_i$) signal content ($\delta_i = (p_i - a_i)/a_i$) was calculated for each simulation point. The set of deviations was subsequently binned by sweep parameter "number of reads" and read state (unmerged and merged). As the resulting bins pooled values across all seven insert lengths, this resulted in 140 estimates per bin (Fig 4). Ignoring insert length variation during binning results in increased within-bin variance, but better reflects what could be expected in real-world operational variation. As a further visual aid, a simple plot of predicted to actual was produced for the sweep slice when insert length was 350 bp, and where the jittered points are coloured by their respective read depth (Fig 5).

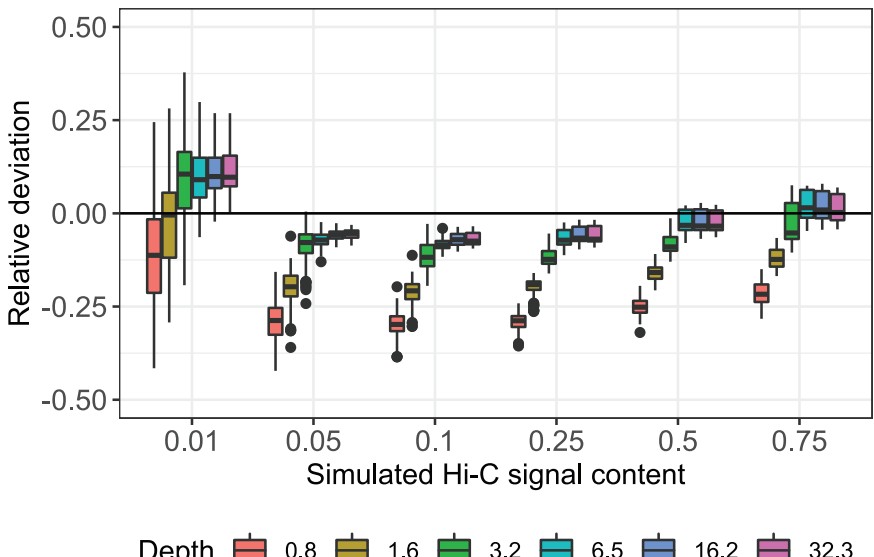

**Fig 4. Relative deviation of predicted from actual with varying insert length.** Relative deviation of predicted from actual Hi-C signal content using qc3C's reference-free method. Number of pairs has been converted to effective read-depth. Each boxplot represents 140 replicates, spanning the simulated range of insert length (150–500 bp). Disagreement at lowest simulated signal content (0.01) is driven by the contribution of the longest insert length (500 bp).

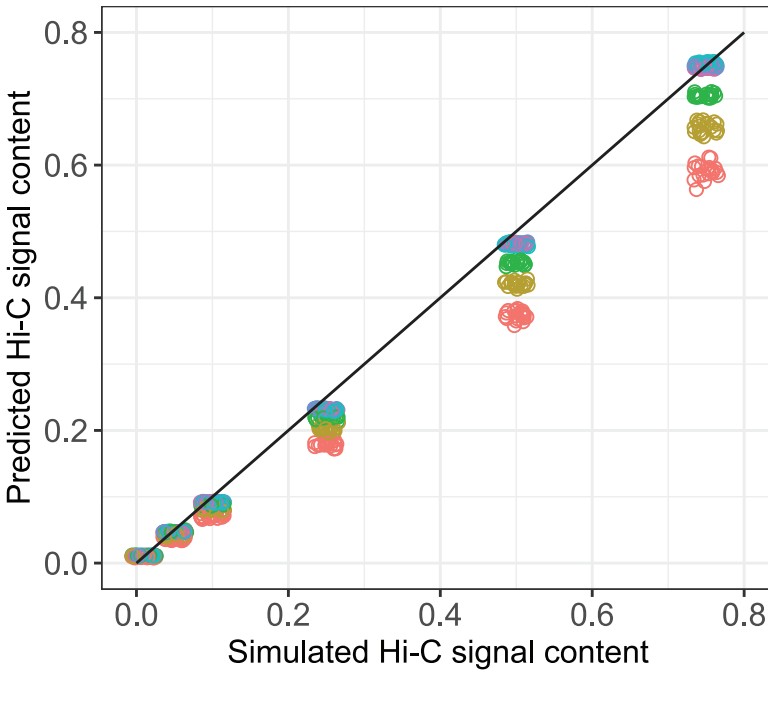

Depth ○ 0.8 ○ 1.6 ○ 3.2 ○ 6.5 ○ 16.2 ○ 32.3

**Fig 5. Predictive accuracy with coverage.** Prediction vs actual, where the line of ideal agreement $y = x$. Holding insert length fixed at 350 bp within the parametric sweep, qc3C reference-free prediction becomes increasingly close to the true value as read-depth increases.

## Real data

A recent comparative analysis considered the impact on Hi-C library quality when taking DNA samples from different tissues (blood, liver) and generating libraries using different protocols (Arima, IconHi-C, Phase), where DNA samples were obtained from *Pelodiscus sinensis* (Chinese soft-shell turtle, PRJNA221645) and human GM12878 (PRJDB8223) [7]. We selected six of the resulting public datasets which differed in their Hi-C protocols (Table 1) and subjected them to fastp clean-up and subsequently to QC analysis with qc3C using both bam and *k*-mer modes (Table 2). Although library insert length would normally be determined through independent means, published values for these libraries are much larger ($\approx 500$ bp) than what are inferred by qc3C bam mode. Having previously encountered libraries whose quoted insert lengths included additional adapter sequence, we elected to supply the *k*-mer mode analysis with insert lengths estimated by the bam mode analysis.

## Discussion

qc3C is the first reference-free method for the assessment of Hi-C library quality, and displays relatively accurate estimation of total Hi-C signal content over widely varying simulated experimental conditions. By eliminating the dependency on reference genomes, qc3C enables quality assessment for any Hi-C project which has reached the stage of sequencing, simplifying the data production workflow. qc3C eases the institution of a quality management practice in which QC performed at the pilot scale informs decision making at full-scale (Fig 6).

**Table 1. Pilot Hi-C sequencing libraries subjected to QC analysis with qc3C.** The two Arima libraries differ in the inclusion of an additional step of T4 polymerase digestion to suppress the population of dangling-ends, while the human GM12878 libraries tested formalin fixation time. All libraries were sequenced on an Illumina HiSeq 1500 instrument.

| Sample | Accession | Organism | Enzyme/Kit | Notes |
|--------|-----------|----------|------------|-------|
| Ps_HindIII | DRR177157 | *P.sinensis* | HindIII | – |
| Ps_DpnII | DRR177159 | *P.sinensis* | DpnII | – |
| Ps_Phase | DRR177161 | *P.sinensis* | Phase | – |
| Ps_Arima | DRR177163 | *P.sinensis* | Arima | T4 minus |
| Ps_T4_Arima | DRR177164 | *P.sinensis* | Arima | T4 plus |
| Hs_10_HindIII | DRR177165 | *H.sapiens* | HindIII | 10 min fixation |
| Hs_30_HindIII | DRR177166 | *H.sapiens* | HindIII | 30 min fixation |

## Simulation results

The performance of our reference-free estimation of Hi-C signal content has been tested using simulated libraries over a wide range of experimental conditions. Considering the relative deviation of prediction to expected over the sweep (Fig 4), it can be seen that our reference-free method tends to underestimate the true signal content in most scenarios. Predictive accuracy and precision, however, improve noticeably as the size of the simulated library increases and close agreement with the expected is obtainable from a modest amount of sequencing coverage.

The smallest simulated library size of 12500 read pairs (0.8-fold coverage) has the highest variance in prediction across all simulated signal levels, while also displaying the greatest tendency for systematic underestimation. Accuracy improves rapidly as simulated library size increases from 12500 to 50000 (0.8 to 3.2-fold coverage), after which there are diminishing returns (3.2 to 32.3 times coverage). It is reasonable to expect that this plateau in performance gain, coming in at approximately 5 times coverage, can be reached by typical real-world sequencing run yields.

The sweep was fully partitioned to inspect the behaviour of our reference-free estimation with change in experimental parameters. Here, each resulting bin contains 20 replicates differing only by random seed (Fig 7). For moderate to high signal levels (signal $\geq$ 0.05), the precision and accuracy of our reference-free method is largely insensitive to changes in insert length. For libraries with the least signal content (signal = 0.01), however, reference-free estimation tends to overestimate at what should be a reliable level of coverage ($\geq$ 3.2-fold coverage) and displays increased sensitivity to change in insert length. This apparent difficulty in prediction at low signal levels might be explained by the drop in statistical power, where for

**Table 2. Quality assessment results from qc3C in both bam and _k_-mer modes.** Bam mode statistics are highlighted in blue, while _k_-mer mode statistics are highlighted in orange. Adjusted read-through has been corrected for the unobserved fraction.

| Sample | Insert length | Unobs frac | *trans*-mapping | Read thru | Pairs >10kb | Adj Read-thru | Hi-C fraction |
|--------|---------------|-----------|-----------------|-----------|-------------|---------------|---------------|
| Ps_HindIII | 217 bp | 3.1 % | 56.1 % | 35.8 % | 28.3 % | 35.8 % | 43.1 % |
| Ps_DpnII | 219 bp | 4.7 % | 58.7 % | 49.1 % | 29.5 % | 49.1 % | 56.5 % |
| Ps_Phase | 511 bp | 54.4 % | 32.0 % | 5.1 % | 6.7 % | 10.9 % | 15.4 % |
| Ps_Arima | 297 bp | 22.8 % | 48.3 % | 30.8 % | 27.3 % | 38.2 % | 56.2 % |
| Ps_T4_Arima | 319 bp | 27.7 % | 50.9 % | 29.9 % | 28.9 % | 39.6 % | 59.4 % |
| Hs_10_HindIII | 248 bp | 7.9 % | 18.0 % | 41.4 % | 68.4 % | 43.2 % | 55.8 % |
| Hs_30_HindIII | 248 bp | 10.0 % | 12.8 % | 36.7 % | 55.2 % | 38.3 % | 51.9 % |

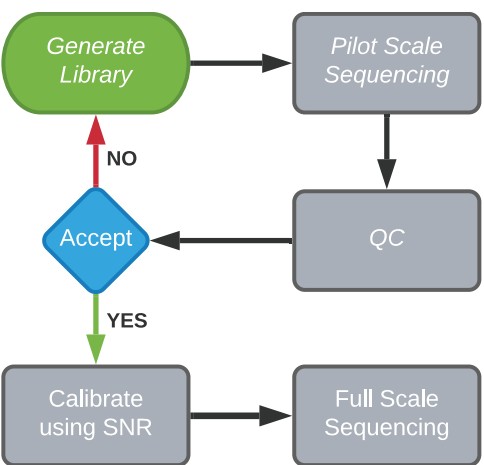

**Fig 6. Mitigation strategy.** A simple QA directed mitigation strategy, where Hi-C sequencing libraries are assessed for quality (SNR) using a small-scale pilot sequencing run. Very poor quality libraries can be rejected, while accepted libraries go on to deeper sequencing. The total required yield of paired reads can be calibrated by the inferred SNR. Given sufficient starting material, Hi-C library generation can be reattempted when previous libraries are abandoned.

50000 read-pairs, a signal level of 0.01 produces roughly 500 proximity ligation containing pairs, of which only a portion are observed. Also at lower coverage, it is more difficult to distinguish a true LA from a naturally occurring sequence. The effect of merging overlapping read-pairs using fastp saw only a slight improvement and only when simulated library insert length was sufficiently small ($\leq$ 300 bp) (S3 and S4 Figs).

For comparison, the tool hicQC (default options) from the HiCExplorer suite [19, 20] was used to infer signal content. Here, signal content was calculated as the ratio of the reported statistic "Hi-C contacts" against the number of reads analyzed. When the sweep is partitioned in equivalence to our reference-free method above (Fig 8), it can be seen that signal content was underestimated for most simulated scenarios, decreasing steadily—for the most part—as insert size increases. The exception to this trend occurs for small insert sizes (150–200 bp) and lower signal levels (0.01–0.25), at which point overestimation can occur. In particular, when insert size was equal to read length (150 bp) and signal content was lowest (0.01), signal content was more than 20-fold overestimated by hicQC. Variation between simulation replicates follows typical counting statistics, decreasing steadily as signal content increases.

Given sufficient coverage ($\geq$ 3.2-fold), our reference-free estimation method compares favourably with the quality metric derived from HiCExplorer quality statistics, obtaining as good or better estimations of signal content. However, it must be remembered that the "Hi-C contacts" statistic reported by hicQC is equivalent to the number of pairs accepted by HiCExplorer when constructing the contact matrix for a given library. As such, used as a measure of signal content, this value should be expected to under-report—relative to the true value—as each pair must pass multiple criteria intended to minimise the false-positive rate (noise) within the matrix.

## Real data results

Using qc3C's bam mode analysis, the impacts of protocol choices noted by [7] are evident and the analysis demonstrates the potential utility of qc3C for troubleshooting library generation and protocol optimisation (Fig 9). For example, libraries Hs_10_HindIII and Hs_30_HindIII

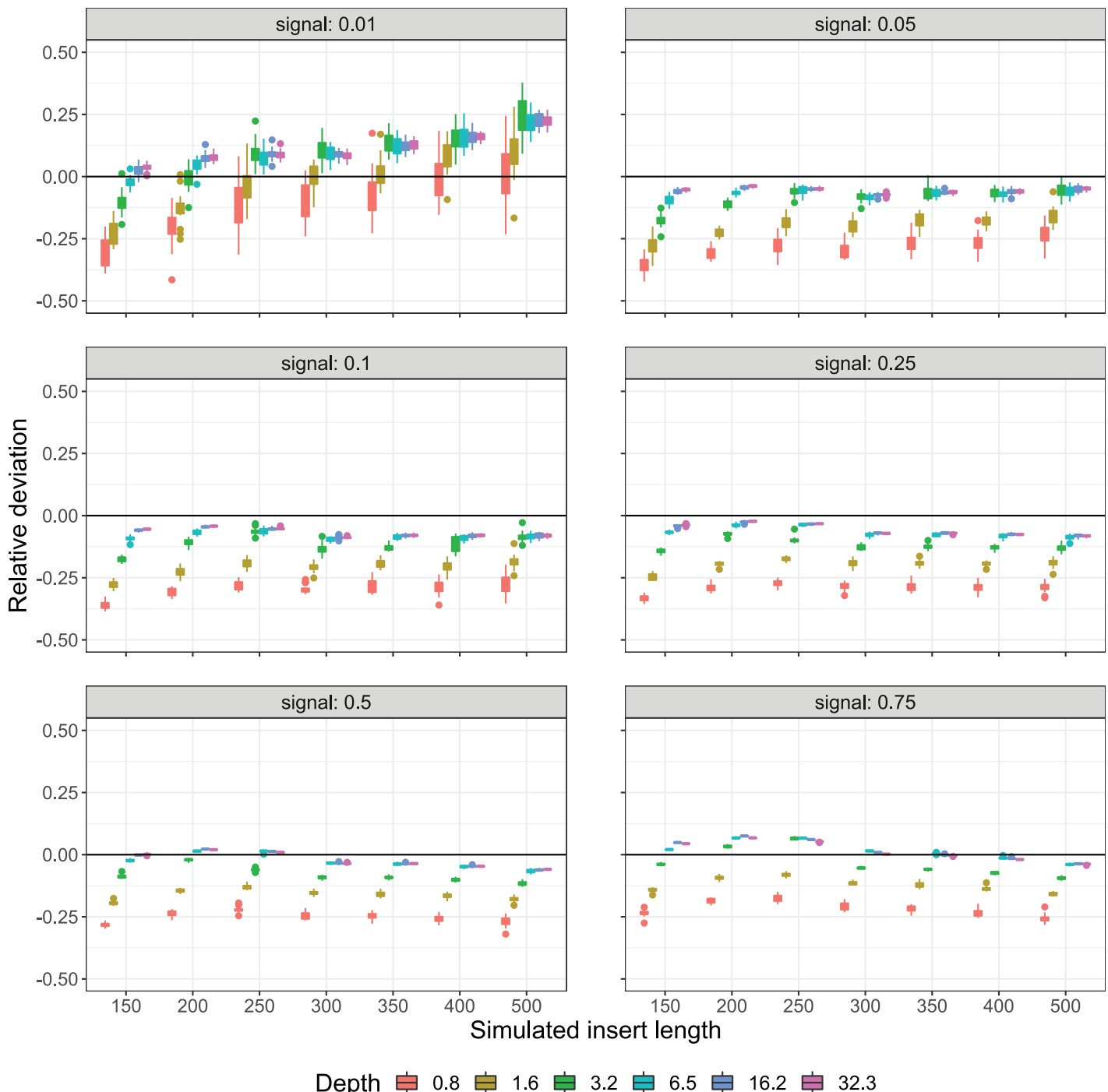

**Fig 7. Relative deviation from actual for simulation replicates.** Deviation of predicted from actual Hi-C signal content, where the simulated sweep has been fully partitioned in each inset plot. Boxes represent the [25:75] quantiles of each set of 20 simulation replicates. A greater variation and sensitivity to insert length at the lowest signal level (0.01) is evident. At higher signal levels the prediction is relatively stable with respect to insert length and intra-replicate variation is small.

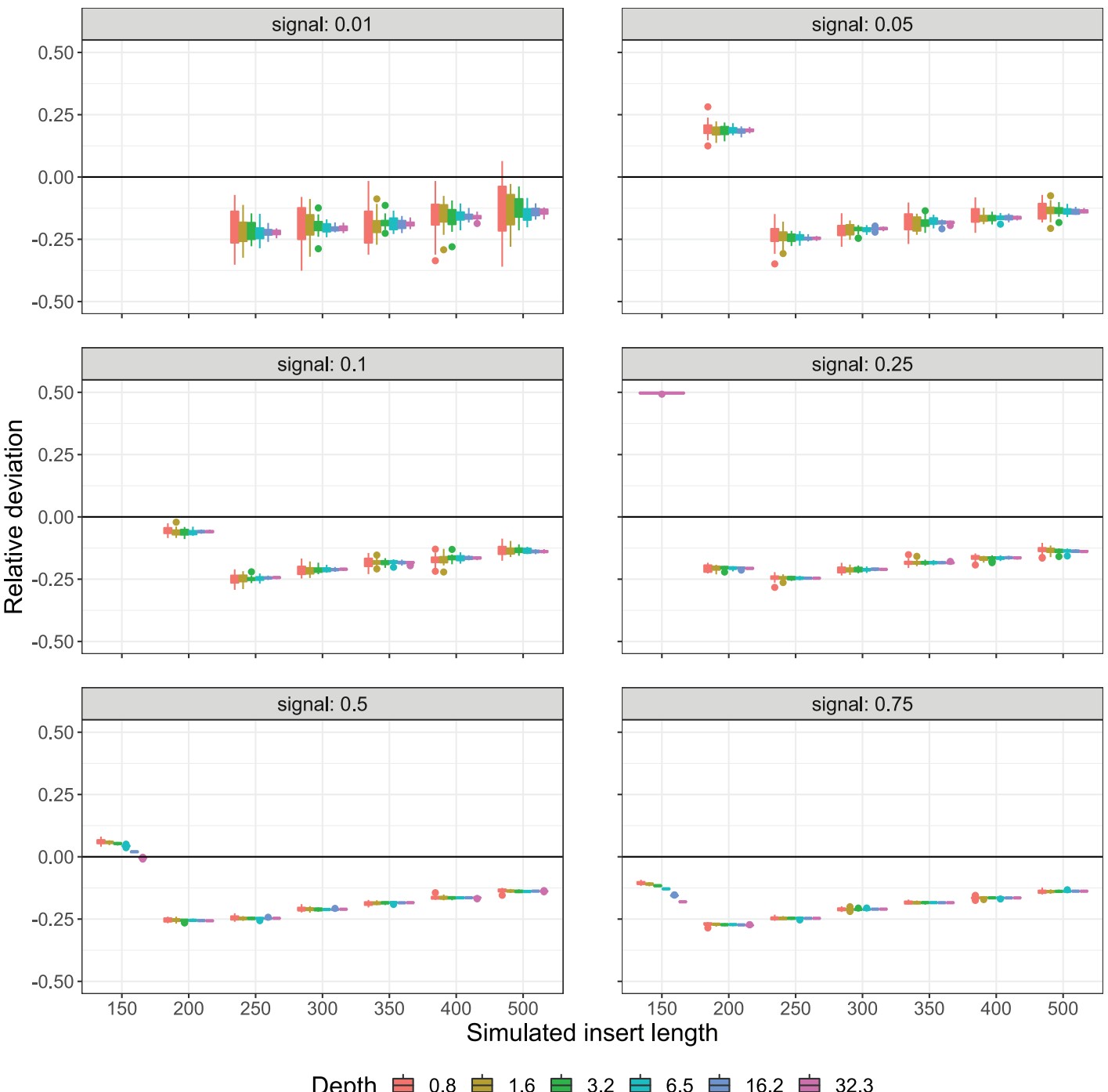

**Fig 8. HiCExplorer contact count comparison.** Signal content was estimated as the number of determined Hi-C contacts relative to the number of analyzed read-pairs using the tool hicQC from HiCExplorer. The deviation of this prediction from the actual fraction of simulated Hi-C read-pairs is shown for the fully partitioned sweep, where all bins contain 20 replicates differing only by random seed (equivalent to Fig 7 and S4 Fig). It can be seen that this statistic leads to consistent underestimation of Hi-C signal content for the majority of simulated scenarios, with the exception of small insert sizes (150–200 bp), which at lower signal levels (0.01–0.25) can result in significant overestimation. Missing points correspond to large relative deviation ($> 0.5$) which fall out of range on the y-axis.

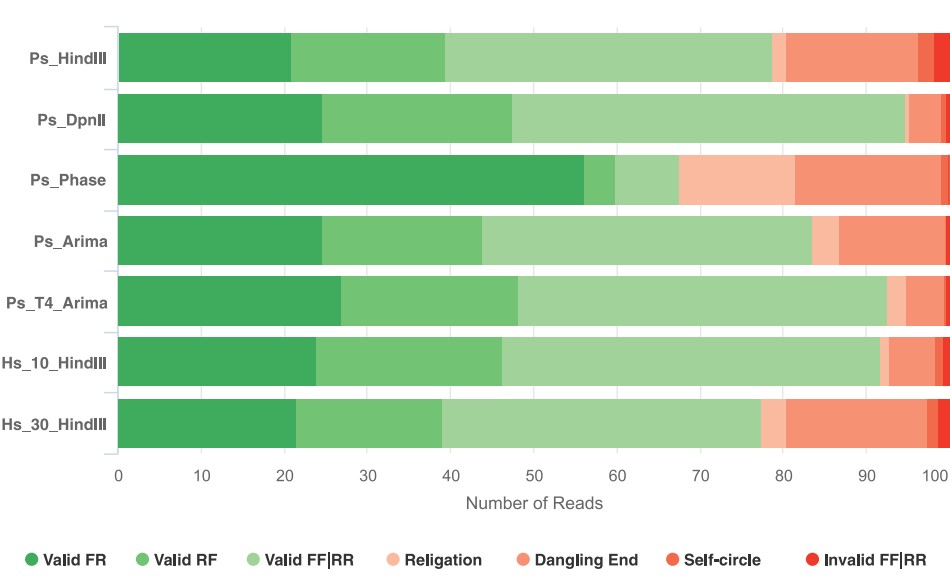

**Fig 9. Valid vs invalid pairs.** The results of qc3C bam mode analysis showing the breakdown of valid and invalid pairs for each library. Valid pair classes are shades of green, while invalid pair classes are shades of red, while two-letter acronyms refer read-pair orientation (F: forward, R: reverse).

differ only in their formalin fixation time (10 and 30 minutes respectively), and it can be seen that increased cross-linking was detrimental and resulted in more than double the proportion of invalid religation and dangling-end pairs. The excessive cross-linking significantly reduced the proportion of read-pairs separated by more than 10kbp (68.4% down to 55.2%) and trans-mapping pairs (18% down to 12.8%). The negative impact can also be detected in qc3C $k$-mer mode, where the estimated Hi-C signal content dropped from 55.8% to 51.9%. In another case, the only difference between libraries Ps_Arima and Ps_T4_Arima is the introduction of a T4 polymerase treatment step meant to reduce the proportion of dangling-ends. The qc3C bam mode analysis indicates this was successful as the proportion of dangling-ends dropped (12.7% down to 4.7%), with a corresponding gain in valid Hi-C pairs. Again, in qc3C $k$-mer mode the change in quality is evident with Hi-C signal content improving from 56.2% to 59.4%.

The distribution of mapped read pair separation is another output of qc3C bam mode analysis (Fig 10). For the seven libraries that were analyzed, most exhibit a similar fall-off in density with increasing separation, with the exception of the Ps_Phase that was created using a shortened duration transposase-based protocol. All libraries produce pairs with over 1 Mbp separation and all possess the characteristic large peak below 1000bp; composed of a mixture of primarily invalid but also short-range valid pairs. Making the assumption that the large peak is dominated by conventional (non-PL) fragments, qc3C calculates library mean insert length using a robust statistical estimator known as sigma-clipping (Table 2).

Of all the statistics provided by qc3C bam mode, the indicator which perhaps best correlates with Hi-C signal content is the proportion of observed read-through events. For a read to qualify, its alignment must end early at a cut-site and the remaining extent must either not align or be split aligned. It is a stringent and explicit test for features expected of Hi-C read-pairs and,

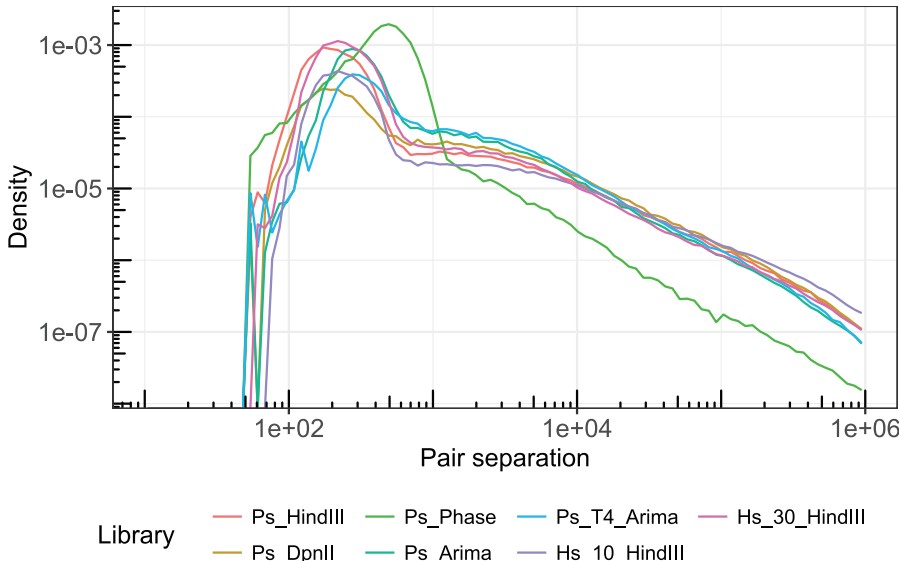

**Fig 10. Distribution of pair separation.** A log-log plot of the distribution of pair separation as provided by qc3C bam mode. The majority of libraries exhibit a similar fall-off with increasing separation, while Ps_Phase falls off more rapidly. For all libraries, the great majority of density falls well below 1000 bp and represents uninformative short-range or invalid pairs.

like our reference-free *k*-mer method, is affected by the unobserved fraction of a library. Once corrected for the unobserved, adjusted read-through follows a similar trend to reference-free estimation (Table 2).

Two of the analyzed libraries derive from human DNA and consequently access to a high quality reference genome sequence. The remaining six that derived from *P. sinensis* DNA had access to a high quality draft genome (2.2Gbp extent in 19,903 scaffolds, N50 3.4Mbp). The impact of this much less refined reference on Hi-C quality statistics can most easily be seen by considering the fraction of inter-contig pairs. For the two human libraries an average of 15.4% pairs spanned contigs, while for the remaining six this rose to half of all pairs (49.2%). Assuming roughly equal interchromosomal interaction rates for human and softshell turtle, then roughly a third of all turtle Hi-C pairs—all of which might represent intra-chromosomal contacts—could not be assessed as such. For libraries which employed the same protocol, but different DNA sources (human: Hs_10_HindIII, turtle: Ps_HindIII), pairs separated by more than 10kbp are more than twice as common in the human library (68.4% vs 28.3%). While differences in the 3D structure of chromatin for distantly related species might influence the fraction of long-range pairs, were it to explain the above observed discrepancy, we would expect to see a similar trend in other statistics such as the fraction of read-through events or our estimation of Hi-C signal content. Though the fraction of read-through events requires a reference genome for calculation, long-range contiguity is not a requirement. It happens that neither the fraction of read-through events nor reference-free estimation of Hi-C fraction supports the discrepancy seen in long-range pairs. Therefore we can surmise that the difference in reference genome quality is at least part of the explanation and demonstrates the potential for a reference-based statistic to be skewed by reference quality.

## Limitations and future directions

The Hi-C read simulator sim3C was chosen for its parametric control over data characteristics, however, it is not capable of correctly simulating Hi-C data for genomes with complex 3D structures such as the human. As such, the simulation sweep was based upon a single bacterial genome, which may have led to unintended simplifications in the analysis.

When the reference is a draft assembly, where the correct assignment of contigs to chromosomes is unknown, true inter-chromosomal pairs and intra-chromosomal pairs mapping to separate segments (contigs) of the same chromosome are indistinguishable. This can skew interpretation of library quality from qc3C's bam mode, if based solely on statistics sensitive to reference quality.

A key assumption in correcting the raw Hi-C fraction is that loci of proximity junctions are distributed uniformly at random across the inserts. If in fact the density is non-uniform, this will introduce a systematic error. In particular, if the density is centralised and more akin to a beta distribution with shape parameters $\alpha = \beta > 1$, then our estimate of the unobserved fraction of Hi-C junctions will be underestimated and lead to an underestimation of the total Hi-C fraction.

For good accuracy and precision with reference-free estimation of Hi-C signal content, qc3C likely requires a Hi-C library with more than 3-times coverage of the most abundant genome in the sample. This is driven by the need to construct a $k$-mer library with reliably sampled frequencies, since it is necessary to judge rare $k$-mers introduced through the Hi-C protocol against the spectrum of those that are naturally occurring. We speculate that Hi-C signal content can be detected using junctions in high copy number regions such as repeat families and mitochondrial DNA even when modal coverage for unique genomic regions is much lower than 3-times. We further speculate that, in cases of particularly low signal estimates, an accompanying shotgun library could optionally be leveraged to estimate this background $k$-mer spectrum.

For both modes of operation, it is important that read data is subjected to clean-up to eliminate sequence contamination (such as adapters). When libraries possess significant overlap in read pairs, we have found improved results by first merging pairs, however at present, qc3C only supports pair merging using the tool fastp. Using our reference-free method, we speculate that filtering and reporting of confident true Hi-C read-pairs is a possibility, but outside the scope of our original goals.

## Supporting information

**S1 Fig. qc3C inputs and outputs.** The two primary modes of operation A) BAM (reference-based) and B) KMER (reference-free) are invoked by the sub-commands `qc3C bam` and `qc3C kmer`. Additionally, the sub-command `qc3C mkdb` is provided as a convenience for constructing Jellyfish $k$-mer libraries. Reference-based BAM mode requires as input the reference in FASTA format and a user-made BAM file containing alignments of a Hi-C paired-end read-set to the supplied reference. Reference-free KMER mode requires as input only a Hi-C paired-end read-set in FASTQ format. If no $k$-mer database is supplied to KMER mode, qc3C will offer to create one, otherwise users can themselves create a database using the `qc3C mkdb` sub-command. The primary modes emit different QC reports, which is written to the console, and JSON and HTML formatted files.
(PDF)

**S2 Fig. qc3C algorithm.** Symbolic workflow of the qc3C algorithm. A) Beginning with a Hi-C read-set, Jellyfish is used to generate a $k$-mer database for a user-selectable size $k$. Afterwards,

observations of relative coverage $\phi_i$ are collected both around ligation artefacts (LA) and randomly for ordinary regions. B) Collected observations of $\phi_i$ for ordinary regions are used to compute empirical ranked p-values. Computed p-values are subsequently distributed to the observations of $\phi_i$ around LA and a final estimation of the fraction of Hi-C read-pairs is obtained as the weighted sum of these p-values. Bootstrapping is used to infer a 95% confidence interval.
(PDF)

**S3 Fig. Relative deviation for merged pairs with varying insert length.** The relative deviation of predicted from actual Hi-C signal content using qc3C's reference-free method when overlapping read-pairs are merged prior to analysis. The sweep has been partitioned so that bins contain 140 sample points of varying insert length (equivalent to Fig 4). For merged read-pairs, there is a small improvement in performance relative to unmerged read-pairs, however prediction at the lowest simulated Hi-C signal level (signal = 0.01) remains the least accurate.
(PDF)

**S4 Fig. Relative deviation for merged pairs replicates.** The sweep has been fully partitioned and all bins contain 20 replicates, differing in random seed only (equivalent to Figs 7 and 8). Relative deviation of predicted from actual Hi-C signal content closely resembles that of unmerged read-pairs. Since reads were simulated at a length of 150 bp, read-pair overlap is most pronounced when insert length was $\leq$ 300 bp. At insert lengths larger than this threshold, predictions become increasingly similar to the unmerged results. Overall, between merged and unmerged read-pairs, there are only slight differences in the mean and variation of replicates at equivalent points in the sweep.
(PDF)

**S1 Supporting Information. Example outputs.** Aside from reporting quality results to the user via the console, an analysis run produces a quality report written to disk in both HTML and JSON formats. The create if either output format can be disabled. The JSON format files can be imported by MultiQC. This zip archive includes example results of both BAM and KMER modes, as well as the resulting MultiQC report.
(ZIP)

## Acknowledgments

We would like to thank Dr. Bas Dutilh for helpful conversations about detecting signal in meta3C data.

## Author Contributions

**Conceptualization:** Matthew Z. DeMaere, Aaron E. Darling.

**Data curation:** Matthew Z. DeMaere.

**Formal analysis:** Matthew Z. DeMaere.

**Funding acquisition:** Aaron E. Darling.

**Investigation:** Matthew Z. DeMaere.

**Methodology:** Matthew Z. DeMaere.

**Project administration:** Matthew Z. DeMaere, Aaron E. Darling.

**Software:** Matthew Z. DeMaere.

**Supervision:** Aaron E. Darling.

**Validation:** Matthew Z. DeMaere.

**Visualization:** Matthew Z. DeMaere.

**Writing – original draft:** Matthew Z. DeMaere.

**Writing – review & editing:** Matthew Z. DeMaere, Aaron E. Darling.

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
