## [Decision Letter · Decision Letter 0]

30 May 2021

Dear Dr DeMaere,

Thank you very much for submitting your manuscript "qc3C: reference-free quality control for Hi-C sequencing data" for consideration at PLOS Computational Biology.

As with all papers reviewed by the journal, your manuscript was reviewed by members of the editorial board and by several independent reviewers. In light of the reviews (below this email), we would like to invite the resubmission of a significantly-revised version that takes into account the reviewers' comments.

We cannot make any decision about publication until we have seen the revised manuscript and your response to the reviewers' comments. Your revised manuscript is also likely to be sent to reviewers for further evaluation.

Sincerely,

Mihaela Pertea

Software Editor

PLOS Computational Biology

Mihaela Pertea

Software Editor

PLOS Computational Biology

Reviewer's Responses to Questions

**Comments to the Authors: **

Reviewer #1: This manuscript presents a new tool for evaluating the quality of Hi-C data, with or without a reference genome. The statistical method seems appropriate and based on reasonable assumptions, and there is currently no other method to satisfactorily assess the quality of a Hi-C library without a reference genome. While traditional sequencing QC measures can be used in the presence of a high-quality reference, qc3C presents an option for QC without a reference, which is likely to become more common as Hi-C is used more outside of humans and traditional model organisms. Some minor changes to improve clarity of the manuscript would be helpful, but otherwise the method is presented well and is likely to be useful to the community.

1. The figures overall are very high-quality and helpful, but it would be great if an extra figure could be added describing the qc3C method itself. It is a bit hard to follow the text alone in the methods section, and a visual aid here would be great. If there is concern over too many figures, I think Figure 1 could be removed (just cite the original Hi-C paper from 2009 which has a very similar figure to describe the protocol). 

2. The method requires an exact sequence match to identify a ligation artefact (LA), and results suggest that qc3C is more likely to underestimate the proportion of valid proximity ligation pairs. Would allowing for indels or sequencing error when counting LAs potentially improve this undercounting? Or is the exact sequence match likely to capture all LAs, and the underestimates are due to other factors?

Minor points for clarity:

1. My personal preference for ease of reading would be to discuss the results of the simulation and real data analysis right after the relevant data are described, rather than splitting these across sections so the reader has to flip back and forth to remember the experimental set-up when reading about the results.

2. There are many acronyms introduced in the paper, and it becomes harder to remember them by the end. Especially for those used in figures, it would be helpful to put the full terms in the caption (ie, FF/RR/FR/RF in Figure 7 refer back to terms used in Figure 2, and are easy to have forgotten).

3. For the real data results, it would be easier to follow if the libraries were renamed to something descriptive/interpretable, with a table mapping these to the official library names (or just add a column with these descriptive names to Table 1). It is very difficult to interpret Figure 8 without flipping through the text multiple times to find the descriptions of each sample to determine which are human and which are turtle, and what the differences are between them all. For example, DRR177157 could be referred to as “P.sinensis_HindIII” both in the text and in the figure, so the reader automatically knows the species and defining characteristics of this library.

Reviewer #2: In this manuscript, authors present a new tool, qc3C, which is able to control the quality of Hi-C libraries without the need of an assembled genome by using the uneven Hi-C libraries k-mers distribution generated by ligation artifacts. This QC toolbox will be of particular interest to researchers wanting to assemble new genomes, or especially metagenomes, using Hi-C libraries. It is also an interesting tool for any Hi-C as an “all-in-one” Hi-C QC toolbox, providing enzyme configurations for Arima and Phase Hi-C kits, which makes it easy to use.

The manuscript is well written, the figures are clean and the tool will be of use to the field. The provided Zenogo repository was greatly appreciated during the review process, as well as the manuscript formatting. Despite the evident quality of the manuscript, major key points are lacking for the manuscript to be readily accepted for publication. 

- The proper figures and analyses to clearly show how the reference-free mode compares to the more precise bam mode to validate the quality of Hi-C libraries. Table 2 is insufficient to that end. For instance, scatter plots displaying “read thru” bam compared to no-reference Hi-C reads fraction, and bam mode valid pairs compared to no-reference Hi-C fraction would help to understand how these different metrics are related to one another for valid and low quality Hi-C libraries. 

- A table showing the currently available Hi-C QC tools with their respective features would help highlighting the qc3C features and what it adds to the field.

- Figure 7 should compare results obtained by qc3C bam mode and Hi-C Pro QC tool for each type of read pairs. Comparing qc3C bam mode to HiCExplorer hicQC is also important to assess how qc3C performs compared to other tools.

- Figure 8 should be changed to show interactions larger than 1Mb range. Biologically relevant interactions are seen at that range. Moreover, monitoring contacts above the 1Mb range between Hi-C libraries help to determine how comparable Hi-C datasets are.

- I was unable to test the tool on a computing cluster. After following the installation process detailed on the github page of the tool, using the bam mode, I received an error concerning “/lib64/libm.so.6: version `GLIBC_2.23' not found”. After installing the tool on my local machine I was able to test it and it seemed to work in bam mode. However I was unable to load separated bam files produced by BWA for HiCExplorer (R1.bam and R2.bam, i.e. non-merged bams). Such a mode would be necessary so that qc3C can be used with any type of Hi-C bam file (splitted or merged).

- The methodology concerning the reference-free mode, while explained rather thoroughly, requires a supporting figure explaining each step so that the method can be better understood. For instance, authors could provide a figure explaining the two methods parts (observed fraction and correction for the unobserved). 

- A figure explaining the inputs and outputs and connexion between the tree CLI tools of qc3C should be provided.

- An example of the outputs of the tools should be provided in the article in a complete figure.

There are also Minor comments to address:

- qc3C should provide the number of duplicated sequences in the multiQC report.

- Figure 2 is cited in the text before Figure 1

- Figure 2 has several issues that need to be corrected. Change Figure 2 Top and Bottom to Figure 2 A and B. Figure 2 Invalid pairs figure is missing some cases of pairs that should be removed from Hi-C datasets, for instance singletons / one mate unmapped reads. Check for instance Lajoie et al. 2015, “The Hitchhiker's Guide to Hi-C Analysis: Practical guidelines” Figure 2, or the HiCExplorer hicQC documentation (https://hicexplorer.readthedocs.io/en/latest/content/tools/hicQC.html). The “religation” invalid pairs on Figure 2 bottom schematic seems to be a valid pair from most common representations of Hi-C reads.

- Long range pairs are much farther than 1000bp, especially since 1kb resolution matrices are common in Hi-C and 1000bp represents two adjacent bins, i.e. the closest possible range available in most Hi-C experiments. Long range contacts are depending on the chromosome scales and genomes. HiCExplorer for instance chooses to separate short and “long” range contacts at the 20kb threshold. But interesting and biologically relevant interactions are seen above the 1Mb range in many organisms. That 1000bp threshold must be changed in the text or specified that it concerns genomes with very small chromosomes. Also, it is advised to think in terms of gradual genomic separation and ranges rather than the binary short vs long range interactions - that is precisely what Distance vs Counts corresponds to (number of contacts in Y, log10 genomic separation in X, represented in Figure 8) and it is a much better QC assessment than proportion of long range vs short range interactions.

- Authors need to describe more in-depth what is the expected frequency of reads displaying the sequenced ligation junctions (line 150). Dangling ends should most likely display the ligation junction sequence while most valid pairs shouldn’t display the ligation junction (depending on the library fragment size and reads length).

- Each part of “Observed fraction” should be splitted in sub parts with subtitles (k-mer extraction, bootstrap...) to improve the comprehension of the method. It would also support the required new associated figure (Method section starting line 154).

- How n = 16 has been determined is unclear (line 159). The size of the random set of paired reads isn’t defined (line 160).

- How many bootstraps are performed on the table T of LA presence/absence per read to assess the confidence interval (line 178)?

- “Reduced to a unique set of values” is not clear (line 182).

- The origin of the p-values is not clear (line 183).

- On which data the bootstrap is performed is not clear.

- Are Merged and unmerged read pairs (line 209-210) corresponding to overlapping and non-overlapping read pairs sequenced from the same fragment (line 200)? The nomenclature should be the same in the whole manuscript.

- What are generated pairs corresponding to (line 229)? The number of simulated Hi-C reads?

- Authors should compare the results of the qc3C reference-free mode, qc3C reference mode, and already published Hi-C QC tools (most likely reference-based) like HiCExplorer hicQC module so that the readers can assess how precise qc3C reference-free and reference-based reports are compared to the tools that already published.

- qc3c --help should display what each available commands (mkdb, bam and kmer) are doing.

- qc3c bam mode: -q MIN_MAPQ, --min-mapq MIN_MAPQ. Minimum acceptable mapping quality [60]. The default value of 60 is potentially dangerous as a default parameter depending on the mapping tool used to generate the BAM file.

- qc3c bam mode: -b parameter documentation is not clear about the possibility to use multiple bam files.

- qc3c bam mode: the check for bam file index should be performed before any computation.

**Have the authors made all data and (if applicable) computational code underlying the findings in their manuscript fully available?**

Reviewer #1: Yes

Reviewer #2: Yes

PLOS authors have the option to publish the peer review history of their article (what does this mean?). If published, this will include your full peer review and any attached files.

Reviewer #1: No

Reviewer #2: **Yes: **Gautier Richard
---

## [Decision Letter · Decision Letter 1]

16 Sep 2021

Dear Dr DeMaere,

We are pleased to inform you that your manuscript 'qc3C: reference-free quality control for Hi-C sequencing data' has been provisionally accepted for publication in PLOS Computational Biology.

Best regards,

Mihaela Pertea

Software Editor

PLOS Computational Biology

Jason A. Papin

Editor-in-Chief

PLOS Computational Biology

Reviewer's Responses to Questions

**Comments to the Authors:**

Reviewer #1: The authors have satisfactorily addressed my previous comments, and I recommend this version for publication.

Reviewer #2: Authors brought the necessary changes to the manuscript or answered comments with detailed arguments, and I thank them for their thorough review of the manuscript. I thus recommend the manuscript to be accepted for publication.

Here are few minor elements to that authors might want to pay attention to during the proof reading of the manuscript:

Figure 3: LA (ligation artifact) could be shown above the red block.

Line 240: unlikely TO be composed entirely OF. "to" and "of" must be added.

Line 241: simultaneous handling OF. "of" must be added.

Figure 4: title is missing a word after actual ?

Table 2: first column should likely be "Sample"

**Have the authors made all data and (if applicable) computational code underlying the findings in their manuscript fully available?**

Reviewer #1: Yes

Reviewer #2: Yes

PLOS authors have the option to publish the peer review history of their article (what does this mean?). If published, this will include your full peer review and any attached files.

Reviewer #1: No

Reviewer #2: **Yes: **Gautier RICHARD

---

## [Editor Report · Acceptance letter]

27 Sep 2021

PCOMPBIOL-D-21-00388R1 

qc3C: reference-free quality control for Hi-C sequencing data

Dear Dr DeMaere,

I am pleased to inform you that your manuscript has been formally accepted for publication in PLOS Computational Biology. Your manuscript is now with our production department and you will be notified of the publication date in due course.

With kind regards,

Amy Kiss
